META-RESEARCH ARTICLE

# Screening, sorting, and the feedback cycles that imperil peer review

**Carl T. Bergstrom**[1]*, **Kevin Gross**[2]*

1 Department of Biology, University of Washington, Seattle, Washington, United States of America,
2 Department of Statistics, North Carolina State University, Raleigh, North Carolina, United States of America

* cbergst@uw.edu (CTB); krgross@ncsu.edu (KG)

## Abstract

Scholarly journals rely on peer review to identify the science most worthy of publication. Yet finding willing and qualified reviewers to evaluate manuscripts has become an increasingly challenging task, possibly even threatening the long-term viability of peer review as an institution. What can or should be done to salvage it? Here, we develop mathematical models to reveal the intricate interactions among incentives faced by authors, reviewers, and readers in their endeavors to identify the best science. Two facets are particularly salient. First, peer review partially reveals authors' private sense of their work's quality through their decisions of where to send their manuscripts. Second, journals' reliance on traditionally unpaid and largely unrewarded review labor deprives them of a standard market mechanism—wages—to recruit additional reviewers when review labor is in short supply. We highlight a resulting feedback loop that threatens to overwhelm the peer review system: (1) an increase in submissions overtaxes the pool of suitable peer reviewers; (2) the accuracy of review drops because journals must either solicit assistance from less qualified reviewers or ask current reviewers to do more; (3) as review accuracy drops, submissions further increase as more authors try their luck at venues that might otherwise be a stretch. We illustrate how this cycle is propelled by the increasing emphasis on high-impact publications, the proliferation of journals, and competition among these journals for peer reviews. Finally, we suggest interventions that could slow or even reverse this cycle of peer-review meltdown.

## Introduction

When we think about the institution of peer review in science, we often envision reviewers as acting akin to shell collectors, sorting through the fragments of shells on a tropical beach in search of whole specimens worth taking home. Indeed, this is part of what peer review does. But papers don't simply appear on *PLOS Biology*'s editorial desk the way that shells wash up on an expanse of white sand. Authors choose to put them there—and they make those choices in anticipation of the evaluation

**Data availability statement:** R code for figures is available at Zenodo, https://zenodo.org/records/15866736.

**Funding:** This work was partially supported by NSF (www.nsf.gov) awards SES-2346645 to CTB and SES-2346644 to KG, by Templeton World Charity Foundation (www.templetonworldcharity.org) Diverse Intelligences frameworks grant 32581 to CTB, and through visitor support at IAST (www.iast.fr) to KG via funding from the French National Research Agency (ANR) under grant ANR-17-EURE-0010 (Investissements d'Avenir program). No funder played any role in the study design, analysis, decision to publish, or preparation of the manuscript.

**Competing interests:** The authors have declared that no competing interests exist.

to follow. Given the stringent review process and the wasted time and effort involved in submitting a paper that is eventually rejected, authors screen their own work, targeting appropriate journals rather than sending everything to the most prestigious outlets. Thus peer review also induces authors to reveal, through their submission decisions, their own private information about the quality of their work [1–3].

However, this service depends on a massive supply of free labor, namely the unpaid and largely uncredited efforts of peer reviewers [4]. But the pool of peer-review labor has become stretched thin, and the problem seems to be worsening. Editors report increasing difficulty in finding reviewers for the manuscripts that they handle. Bibliometric studies in a range of fields support their assertions: reviewers are more likely to decline review invitations and thus the average number of solicitations per acceptance has increased ([5–9], though see [10] for an exception). We appear to be in the midst of a "peer-review meltdown" in which the peer-review system is becoming woefully overtaxed by the volume of manuscript submissions [11–16].

As peer review teeters, scientists have begun experimenting with new ideas to reduce the review load or increase review supply. For example, venues such as Publons and Elsevier's Reviewer Recognition Platform attempt to make reviewing more prestigious by awarding accolades to top reviewers [17,18]. Brokerage services tried charging authors to secure peer reviews that could be forwarded to prospective publication outlets [19,20]. These foundered, but a new generation of journal-independent review initiatives such as Review Commons and Peer Community In have emerged in their stead. Some computer science conferences such as NeurIPS [21] keep review loads down by disallowing revision and re-review. Elsewhere, some journals—including *PLOS Biology* [15]—offer portable peer review, where initial reviews follow a manuscript if it is resubmitted to other venues. Others have suggested tying the opportunity to submit a paper as an author to one's contributions as a reviewer [22], and some journals have experimented with cash payments for reviews [23–26]. Yet others have studied whether machine review using AI [27] and large language models [28] can complement peer review. While debate about the propriety of machine review remains unsettled [29,30], some reviewers are using LLMs for assistance even when journal or conference guidelines forbid doing so [31–33]. Some critics have even proposed eliminating prepublication peer-review altogether [34]. The breadth of these endeavors testifies to scientists' eagerness to place scientific publishing on more stable footing.

Yet all these initiatives are hampered by the fact that a rigorous theory of the structure and function of peer review has yet to coalesce [35]. This article aims to begin to fill that gap. While the literature is dotted with mathematical models of peer review, many of these efforts use detailed, agent-based simulation models that embrace the richness of the scientific ecosystem (e.g., [36–40]). In this paper, we take a different approach by developing low-dimensional models that isolate how peer review shifts the burden of identifying the best science among among authors, reviewers, and readers of the scientific literature. (Zhang et al. [41] have recently presented a model of computer-science conferences that also examines how self-screening by authors affects the peer-review burden.) These models reveal a set of hidden pressures

on peer review, and bringing them to light helps us to understand the tensions that threaten the entire institution as science changes. In particular, these models highlight a pernicious feedback loop: increasing submissions to top journals exhausts the pool of suitable peer reviewers, resulting in lower quality peer reviews that encourage yet more authors to take a chance on submitting their paper to a prestigious venue (Fig 1). We also show how other systemic factors—increasing emphasis on high-impact publications, the proliferation of journals, and competition among these journals for unpaid peer-review labor—propel this cycle. Finally, we consider possible solutions that could interrupt the meltdown of peer review, or even reverse it.

## Model and welfare

### Adda-Ottaviani model

Our model builds from the base model in Adda & Ottaviani (2024; henceforth AO) [42]. AO use their model to study grant competitions, but their model adapts naturally to scientific publication. Our analysis differs substantially from AO, as suits the different aims of the papers. The S1 Appendix provides mathematical proofs of several of the key claims and some additional results.

Consider a scientific community served by two journals: an elite journal that seeks to publish top manuscripts and a mega-journal that publishes everything else. We consider only two journals for the sake of simplicity, although our model applies in any setting with several vertically differentiated journals. Suppose that the elite journal has the capacity to publish a proportion $k \in (0, 1)$ of the manuscripts that the community produces. We assume that the journal's capacity $k$ is

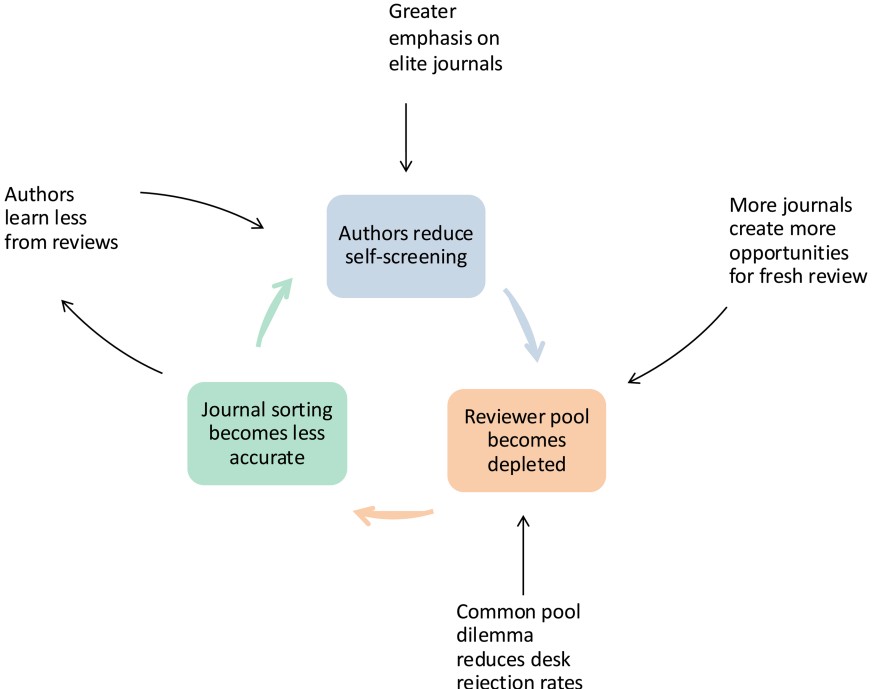

**Fig 1**. **The peer-review meltdown cycle.** Author screening and journal sorting interact in a feedback loop in which inaccurate sorting loosens author screening [42] and looser screening makes sorting less accurate by depleting the pool of available review labor. Forces that exacerbate this feedback loop include (clockwise from top): Greater rewards to publishing lead more authors to submit their paper to top journals; a proliferation of journals gives authors more opportunities to obtain fresh reviews of already rejected manuscripts; journals' reliance on a shared pool of review labor compels journals to underuse desk rejection and overexploit the review pool; and noisier review reduces what authors can learn from having their papers rejected.

determined exogeneously, perhaps by limits imposed by the publisher or by constraints on readers' attention [43]. Henceforth, we focus on the behavior of the elite journal and place the mega-journal in the background. Thus we refer to the elite journal as simply "the journal"; we say that authors who publish their manuscripts in the elite journal are "published", and so on.

Suppose that this community contains a unit mass of authors and that each author is endowed with a manuscript with quality $\theta$. (By a "unit mass", we mean that the community is large enough that we can describe the model in terms of continuously varying proportions of authors. This allows us to sidestep complications that would arise from accounting for the exact size of the community.) For mathematical convenience, assume that $\theta$ has a standard Gaussian distribution across authors, $\theta \sim N(0, 1)$. Authors have their own sense of whether their work is any good. We instantiate this by assuming that an author with a manuscript of quality $\theta$ obtains a private sense of their manuscript's quality $X$ that is drawn from a normal distribution with mean $\theta$ and variance $\sigma_X^2$. Across authors, $X$ is marginally distributed as $N(0, 1 + \sigma_X^2)$. We refer to the quantile $q = F_X(x)$ as the author's type (here and throughout, $F$ denotes a cumulative distribution function, or cdf), and identify authors with their type, e.g. "author $q$".

Authors can submit their manuscript to the journal, or not. Authors who submit their manuscript pay a disutility cost $c > 0$ [3], which includes the opportunity cost of foregoing immediate publication in the mega-journal. Authors whose manuscripts are published receive kudos, prestige, professional rewards, etc. with value $v > c$. More precisely, $v$ gives the additional reward that the author receives from publishing in the elite journal right away relative to the time-discounted reward of publishing in the mega-journal later. We ignore any other actions that authors may take (e.g., cover letters) that could communicate information about their type to the journal.

Because the journal cannot directly observe manuscript quality $\theta$, it solicits reviews in the usual way. For now, assume that journals send out every manuscript they receive for review; desk rejection will be considered later. Let $Y$ be the review score for a submitted manuscript, and assume that for a manuscript of quality $\theta$, $Y$ is drawn from a Gaussian distribution with mean $\theta$ and variance $\sigma_Y^2$. Higher review scores $Y$ provide evidence of higher article quality $\theta$, and thus the journal rationally publishes those papers whose review score exceeds some acceptance threshold $y$.

Two conditions determine the model equilibrium. First, authors submit their paper if and only if their payoff from doing so is positive. We call this the *author-rationality* condition. (Authors on the razor's edge of indifference—those for whom the benefit from submitting their manuscript exactly matches the cost—are rare enough that their behavior does not affect the model equilibrium.) Second, the journal fills its capacity, a condition we call the *capacity-filling* condition. The capacity-filling condition can be motivated by assuming that the journal editor prefers to publish as many papers as possible without exceeding the journal's capacity [44]. See ref. [41] for a related model in which the elite journal is not capacity-limited but instead seeks to publish all papers with a quality above a particular threshold.

Write author $q$'s probability of having their manuscript accepted when facing journal threshold $y$ as $a(q; y) = \Pr\{Y \geq y | F_X(X) = q\}$. (The conditional distribution of $Y$ given $X = x$ is Gaussian with mean $x/(1 + \sigma_X^2)$ and variance $(\sigma_X^2 + \sigma_Y^2 + \sigma_X^2 \sigma_Y^2)/(1 + \sigma_X^2)$.) Because an author's acceptance probability strictly increases with $q$, there is a marginal author $\tilde{q}$ such that only authors with type $q \geq \tilde{q}$ submit their manuscript. A model equilibrium is described by a marginal author $\hat{q}$ and a journal cutoff $\hat{y}$ at which the author-rationality and capacity-filling conditions hold. Writing equilibrium acceptance probabilities as $\hat{a}(q) = a(q; \hat{y})$, the equilibrium solves the author-rationality condition

$$v\,\hat{a}(\hat{q}) = c \tag{AR}$$

and the capacity-filling condition

$$\int_{\hat{q}}^{1} \hat{a}(q)\,dq = k. \tag{CF}$$

Note that *v* and *c* affect the model only through their ratio. AO show that the model equilibrium exists, is unique, and is stable. Fig 2A illustrates the equilibrium's construction.

## Welfare

The scientific literature serves (at least) three different constituencies. *Authors* wish to have their work read. *Readers*—and by extension, publishers who market journals to those readers—want to read about the best and most interesting science, and to do so in a timely fashion. *Reviewers* contribute the volunteer labor that supports the peer-review process. These constituencies represent roles rather than distinct groups; a given researcher may submit a paper on Monday, read several articles on Tuesday, and write a peer review on Wednesday. But by treating each role as a constituency, we obtain a finer-grained view of how peer review trades off benefits across these activities. We define payoffs for each group as follows.

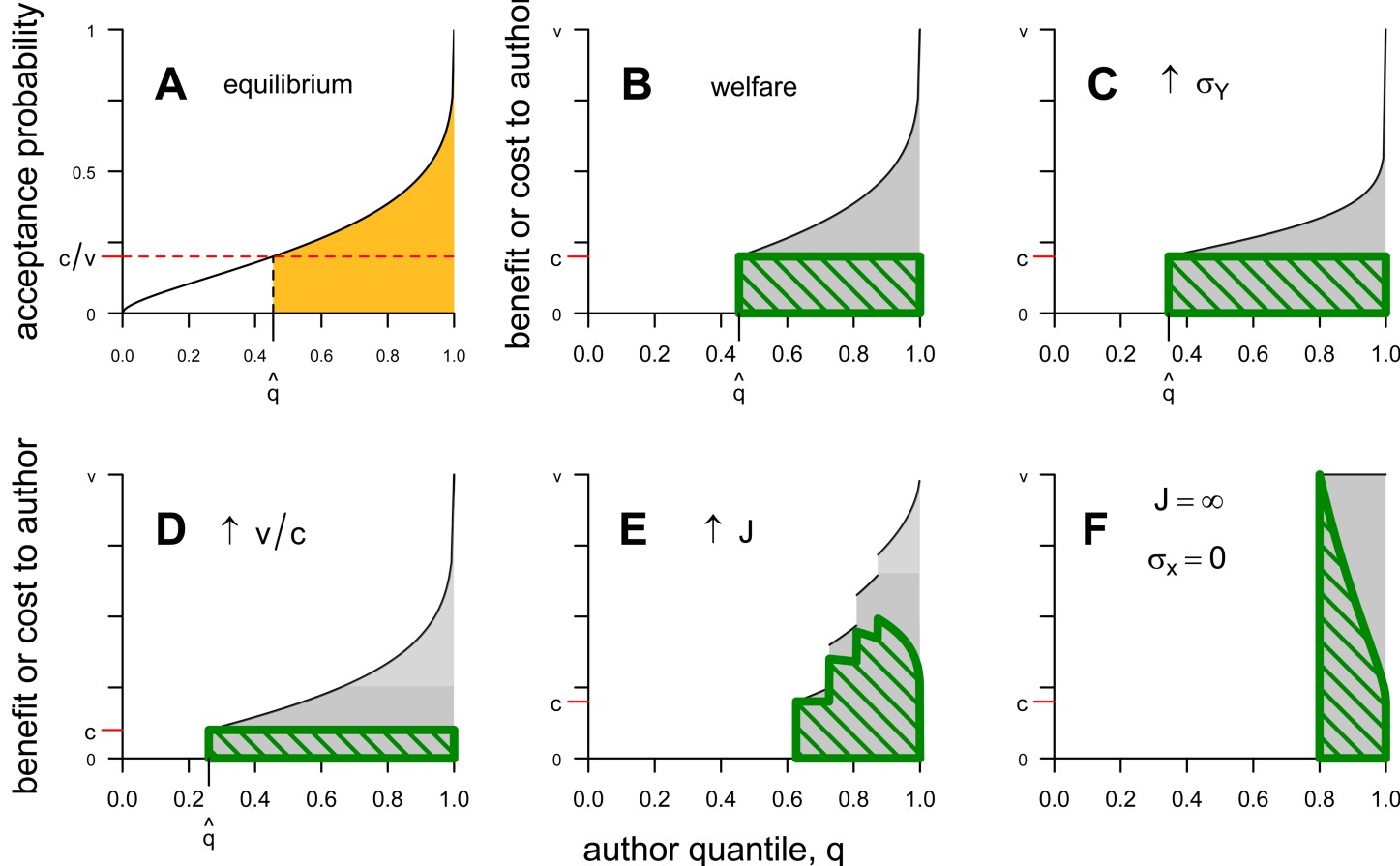

**Fig 2**. **Graphical illustration of the base model of Adda and Ottaviani [42] and associated welfare measures.** A: Equilibrium conditions. The author-rationality condition dictates that the marginal author $\hat{q}$ has acceptance probability $c/v$, and the capacity-filling condition dictates that the area of the gold region—equal to the volume of accepted manuscripts—must equal the journal capacity $k$. Panel based on AO's Figs 2 and 3 [42]. For this panel, $k = 0.2$, $v/c = 5$, and $\sigma_X = \sigma_Y = 1$. B: Welfare measures for the same setting as panel (A). See text for details. C: Change in welfare under greater peer-review noise (in this case, $\sigma_Y = 2$; all other parameter settings as in panel B). D: Change in welfare as $v/c$ increases (in this case, $v/c = 10$; all other parameter settings as in panel B). E: $J = 4$ elite journals. Jumps in the cost and benefit curves show the locations of marginal authors $\hat{q}_i$, $i = 1, \ldots, 4$. F: An infinity of elite microjournals and perfect author knowledge of their manuscript's quality. Code to generate this figure can be found in https://zenodo.org/records/15866736.

**Authors.** Author $q$'s payoff is $v\hat{a}(q) - c$ if $q > \hat{q}$, and 0 otherwise. To measure the authors' welfare, we first sum payoffs across authors to give $\int_{\hat{q}}^{1} [v\hat{a}(q) - c]\, dq = vk - cL$, where $L = 1 - \hat{q}$ gives the volume of papers submitted to the journal, or (in the absence of desk-rejection) the review load. This aggregate payoff simply equals the total rewards from publication, $vk$, minus the total disutility cost paid by authors, $cL$. To obtain a scale-free measure of welfare, we standardize this aggregate payoff by $vk$, the total payoff available to authors when manuscript quality $\theta$ is public knowledge. This yields

$$\frac{vk - cL}{vk} = 1 - \frac{cL}{vk}$$

as a measure of the authors' welfare.

**Readers.** Readers want to read—and journals want to publish—the best manuscripts. To this extent, readers' welfare and the journal's payoff are the same, and we measure them by the average quality $\theta$ of published manuscripts (which can be written as $\mathrm{E}\left[\theta | q \geq \hat{q}, Y \geq \hat{y}\right]$) standardized by the average quality of the best $k$ manuscripts ($\mathrm{E}\left[\theta | \theta \geq F_{\theta}^{-1}(1 - k)\right]$). This measure captures how reliably the journal is able to fill its pages with the best science. As a measure of reader welfare, it is admittedly incomplete because it ignores the speed with which journals disseminate results; all else equal, readers prefer to learn about new discoveries promptly. We lack a metric for reader welfare that incorporates timeliness.

**Reviewers.** While reviewing a manuscript brings a reviewer both benefits and costs, we assume here that the costs of reviewing exceed the benefits. Thus the burden on the review community scales with $L$, the review load.

Author and reviewer welfare can be understood graphically (Fig 2B). The black curve of Fig 2B gives authors' benefit (for the same parameter values as Fig 2A), $v\hat{a}(q)$, and the area under this curve, shaded in gray, equals the authors' total benefit, $vk$. The green hatched region has height $c$ and width $L$, and thus its area equals the authors' total cost, $cL$. Author welfare equals the proportion of the gray shaded area that lies outside the hatched green box. The load on reviewers equals the area of the green box divided by $c$. Reader and journal welfare depends on the actual article quality $\theta$ and thus is determined by the combined action of screening and sorting (Fig 3). In S1 Fig, we show that the optimal strength of screening for the journal and its readers depends on authors' and reviewers' accuracy in assessing manuscript quality. These results are intuitive: journals and their readers are better off with more stringent screening and more selective submissions when authors are well-informed about their paper's quality, and conversely are better off with looser screening and more submissions when referees are better informed.

## Analysis

We begin by analyzing the model with a single elite journal, and then consider a model with several elite journals. Numerical analysis was conducted using R [45].

### A single journal

A central result of AO is that more authors submit their paper when reviewing is less accurate, and vice versa (Fig 2C). The intuition is that less accurate peer review introduces more stochasticity into the journal's sorting. This loosens the relationship between the author's acceptance probability and their type, thus encouraging more authors to submit their manuscripts and relaxing screening. As a result, less accurate reviewing makes both authors and reviewers worse off by increasing the review load $L$. The effect on readers' welfare is ambiguous.

If increasing the review load also decreases the accuracy of peer review, then author screening and journal sorting participate in a feedback loop [41]. It is easy to see how an increase in the review load might cause peer review to become less accurate. Suppose that reviewers differ in their review accuracy and that editors preferentially invite more accurate reviewers. As the review load increases, editors must either ask their favored reviewers to do more work or they must reach further into the pool of reviewers. In either case, peer reviews become less accurate, making the journal's sorting less precise. Thus, review noise $\sigma_Y$ and review load $L$ reinforce each other: more accurate journal sorting compels authors

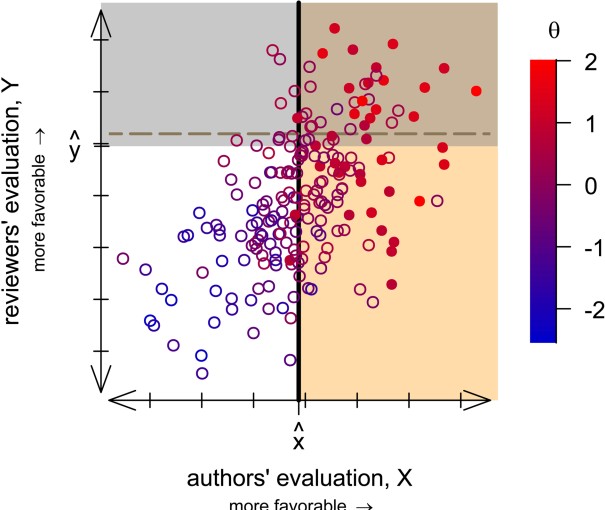

**Fig 3**. **Author screening and journal sorting combine to determine how capably the journal identifies the most publication-worthy science.**
Points show a random sample of 200 manuscripts, located according to the author's sense of the paper's strength $X$ and by a reviewer's score $Y$. Point color indicates the quality of the manuscript $\theta$, with redder (bluer) values corresponding to higher- (lower-) quality manuscripts. The journal publishes $k = 20\%$ of the papers, and filled symbols show the actual top 20% of manuscripts. At equilibrium, only authors who think their paper is at least as good as $\hat{x}$ (tan region) submit their paper, and the journal then accepts papers that reviewers rate as at least as good as $\hat{y}$ (intersection of tan and gray regions). The dashed horizontal line shows the acceptance threshold that the journal would use instead if every author submitted their paper. This figure uses $\sigma_x = 0.5$, $\sigma_Y = 1$, and $v/c = 5$. Code to generate this figure can be found in https://zenodo.org/records/15866736.

to screen more selectively, and tighter screening results in fewer submitted manuscripts and more accurate peer review. In the other direction, less accurate journal sorting results in looser screening, which in turn creates more work for reviewers, leading to even noisier reviewing. (A formal mathematical statement of a model with this feedback loop appears in the S1 Appendix.)

The feedback between screening and sorting affects how the equilibrium changes when the environment is perturbed. For example, in recent decades the career rewards from publishing in top-tier journals have increased dramatically [46, 47]. It is easy to show that when publishing in top journals brings greater rewards—that is, $v/c$ increases—more authors throw their hat in the ring and the review load increases (Fig 2D). The intuition is obvious: when publication is worth more, authors are willing to take a chance on a lower probability of success.

The increase in the review load caused by an increase in $v/c$ is exacerbated by the feedback loop between screening and sorting. Fig 4A,B illustrates by showing how the review load $L$ (determined by the strength of screening) and the review noise $\sigma_Y$ (which determines the accuracy of sorting) shape each other. (Fig 4C will be considered later.) In each panel of Fig 4, the red, blue, and black curves show how the review load $L$ responds to a change in the review noise $\sigma_Y$, for different values of $v/c$. The dashed green curve shows how $\sigma_Y$ might respond to $L$. The equilibrium for a particular value of $v/c$ is found at the intersection of the green curve with the respective red, blue, or black curves. When review accuracy declines with increasing $L$ (Fig 4B), an increase in $v/c$ increases the review load more than it would if reviewer accuracy was independent of $L$ (Fig 4A). In other words, not only does an increase in $v/c$ compel more authors to submit their paper, the resulting pressure on the review pool decreases the accuracy of review, resulting in yet more authors submitting their manuscripts.

The feedback loop between screening and sorting also prolongs transient perturbations to the equilibrium caused by one-time shocks. Although a formal dynamical model is outside the present scope, the intuition for the effect is easily seen. Suppose there is a one-time shock that relaxes screening and increases the review load to a value $L'$ above its

PLOS Biology

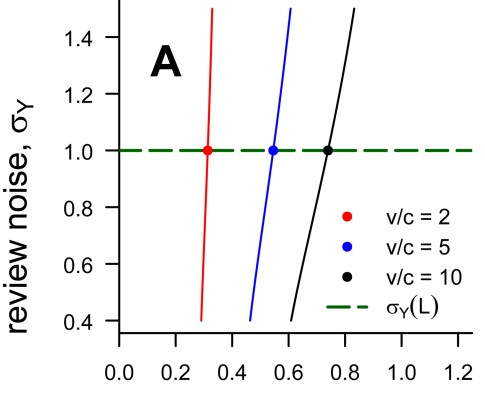
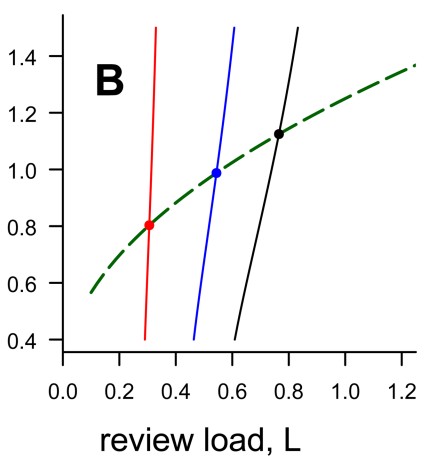
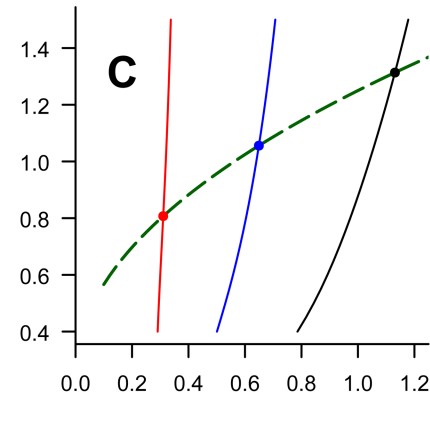

**Fig 4. Feedback effects on peer-review load.** The red, blue, and black curves show the review load $L$ induced by the review noise $\sigma_Y$ for three values of $v/c$. The dashed green curve shows how the review noise $\sigma_Y$ may respond to the review load $L$. Equilibria are found at the intersections of the red / blue / black curves with the green curve. A: A single elite journal and review noise $\sigma_Y$ that is independent of load $L$. B: A single elite journal and $\sigma_Y$ that increases with $L$. C: Three elite journals and $\sigma_Y$ that increases with $L$. Throughout, $\sigma_X = 1$ and $k = 0.2$. Code to generate this figure can be found in https://zenodo.org/records/15866736.

equilibrium level. If the review noise depends on $L$, then the review noise will increase in response to the surge in submissions to a value $\sigma'_Y$. After the shock is over, the submission load does not immediately return to its pre-shock value; instead, it adjusts to the value $L'' < L'$ induced by review noise $\sigma'_Y$. The review noise then adjusts to the value $\sigma''_Y$ induced by $L''$, and so on. Eventually, $L$ and $\sigma_Y$ will return to their pre-shock equilibrium, but these transients will take longer to relax if the review noise temporarily increases in response to the one-time surge in submissions. This suggests one plausible explanation for why a reported change in reviewer behavior following the COVID-19 pandemic [14] may be surprisingly persistent.

## Several competing journals

So far, the model considers a simplified environment with one elite journal and one mega-journal. Of course, the actual scholarly publishing landscape is much richer. As science has grown, the number of journals has increased while (mega-journals aside) the proportion of the community's output published by any single journal has declined [48].

Here, we consider how the shift from a few large journals to several smaller ones has impacted scientists' welfare. We focus on "horizontal" proliferation of journals within a prestige tier. "Vertical" proliferation, in which journals more densely occupy niches along a prestige spectrum, is more difficult to handle because it introduces greater richness into the author's possible actions and consequently lies outside the present scope (but see ref. [44]). We also abstract away from ways in which journals within a prestige tier may specialize to particular subsets of scholarship.

To study the effects of having several elite journals, we embed our previous model in a richer, multi-journal model. Instead of a single elite journal with capacity $k$, now suppose there are $J$ elite journals that are identical in all regards and that each have capacity $k/J$. Extend the model to include an implicit temporal component in which time is divided into discrete periods, and use the previous model to capture authors' and journals' actions within a single period. In each period, the community (though not necessarily the same authors) generates a unit mass of manuscripts with qualities distributed as $N(0, 1)$. At each period, authors who have been rejected fewer than $J$ times can resubmit their manuscript to a different journal. Because the journals are equivalent, authors randomize the order of journals to which they submit their manuscript. Each time an author submits a manuscript, they pay a cost $c$, and they receive a reward $v$ if the manuscript is published. Each time an author is rejected, they rationally become more pessimistic about their manuscript's chances

at other, not-yet-tried journals. To keep the math tractable, we assume that authors only update their beliefs based upon how many times their manuscript has been rejected; they don't observe, or at least don't update their beliefs based upon, the actual review scores $Y$. Journals do not know the submission history of incoming manuscripts and must treat all manuscripts identically.

To develop notation, write the conditional probability that author $q$'s manuscript is accepted on the $j$-th attempt when facing review-score threshold $y$ as

$$a_j(q; y) = \Pr\{Y_j \geq y | F_X(X) = q, Y_1 < y, \dots, Y_{j-1} < y\} \quad j = 1, \dots, J.$$

While $a_j(q; y)$ does not depend directly on $J$, the equilibrium review cut-off $\hat{y}_J$ will depend on $J$, and thus $a_j(q; \hat{y}_J)$ will as well. Write the marginal probability that author $q$'s manuscript is accepted on the $j$-th attempt as

$$b_j(q; y) = \Pr\{Y_1 < y, \dots, Y_{j-1} < y, Y_j \geq y | F_X(X) = q\} \quad j = 1, \dots, J.$$

where $b_j(q; y) = 0$ if it is not worthwhile for author $q$ to submit their manuscript a $j$-th time. Write the probability that author $q$'s manuscript is eventually accepted as $m_J(q; y) = \sum_{j=1}^{J} b_j(q; y)$. The total volume of published manuscripts at each period is then $\int_0^1 m_J(q; y)\, dq$. An equilibrium consists of a review-score cutoff $\hat{y}_J$ at which the journals exactly fill their aggregate capacity and a set of marginal authors $\hat{q}_1 \leq \hat{q}_2 \leq \dots \leq \hat{q}_J$ for which author $\hat{q}_j$ is indifferent about submitting their manuscript for the $j$-th time. The marginal authors satisfy the author-rationality conditions

$$v\, a_j(\hat{q}_j, \hat{y}_J) = c \tag{AR-J}$$

while the capacity-filling condition is given by

$$\int_0^1 m_J(q; \hat{y}_J)\, dq = k. \tag{CF-J}$$

To calculate the review load, write the average number of times that author $q$ submits their manuscript as $\mu_J(q)$, given by

$$\mu_J(q) = \mathbb{1}_{q \geq \hat{q}_1} + \mathbb{1}_{q \geq \hat{q}_2} \times (1 - a_1(q, \hat{y}_J)) + \dots + \mathbb{1}_{q \geq \hat{q}_J} \times \prod_{j=1}^{J-1} (1 - a_j(q, \hat{y}_J)).$$

where $\mathbb{1}_{q \geq \hat{q}_i}$ is an indicator function that equals 1 if $q \geq \hat{q}_i$ and equals 0 otherwise. (Note that $\mu_J(q)$ also gives the aggregate density of incoming submissions across all $J$ journals from type-$q$ authors at each period.) The total volume of submissions, and hence the review load, at each period is then

$$L_J = \int_0^1 \mu_J(q)\, dq.$$

Welfare measures extend naturally. Reader welfare is given by the average quality of published papers standardized by the quality of the top $k$ papers (a formal expression appears in the S1 Appendix). The burden on reviewers is proportional to the review load $L_J$. Author $q$'s payoff equals $v\, m(q) - c\, \mu(q)$; integrating over authors again gives $\int_0^1 [v\, m(q) - c\, \mu(q)]\, dq = vk - cL$; thus standardized author welfare is still $1 - cL/vk$.

The feedback loop between screening and sorting carries forward to this multi-journal model, and the underlying logic remains the same. We focus here on new phenomena caused by the presence of several journals and study a numerical example. This example suggests that increasing the number of journals results in journals publishing better papers while also increasing the review load and the volume of rejected manuscripts. Fig 2E shows how this happens. More journals create more opportunities for authors to have their work considered afresh. More such opportunities increase the volume of recirculated manuscripts, which in turn forces journals to be more selective. Increased selectivity results in higher rejection rates which increases the volume of recirculated manuscripts, and so on. Thus, giving authors more bites at the apple strengthens screening in the sense that fewer authors find it worthwhile to submit their paper in the first place ($\hat{q}_1$ increases with $J$), but because those authors can submit their work several times, the sorting burden grows, and the overall volume of submitted (and rejected) manuscripts rises (Fig 5).

Moreover, increasing the number of journals has a bigger effect on the review load when reviews are noisy (Fig 5). The intuition here is that authors learn less about their manuscript's quality from journal decisions when reviews are noisy than they do when reviews are accurate. Thus, when reviews are noisy, more authors rationally continue to submit the same manuscript despite previous rejections. Combined with the feedback loop between screening and sorting, this identifies yet another feedback cycle by which the burden on peer reviewers grows: as the number of journals proliferates, authors have more opportunities to have their work considered afresh; more such opportunities increase the load on the review community; an increased review load results in less accurate reviews; declining review accuracy diminishes authors' ability to learn, leading more authors to recycle previously rejected manuscripts, and so on. Fig 4C illustrates how an increase in the number of journals amplifies the feedback loop between screening and sorting.

To see the effect of journal proliferation starkly, consider the extreme case in which authors know their manuscript's quality exactly ($\sigma_X = 0$; a similar analysis of this extreme case appears in ref. [41]). In this case, an author's acceptance probability at any given attempt is independent of the number of times the manuscript has already been rejected ($a_j(q; y) = a(q; y)$ for all $j$). Thus there is a single marginal author, and an author's eventual probability of acceptance is $m_J(q; y) = 1 - (1 - a(q; y))^J$. In the limit as $J$ grows large, any paper that worth submitting gets published eventually

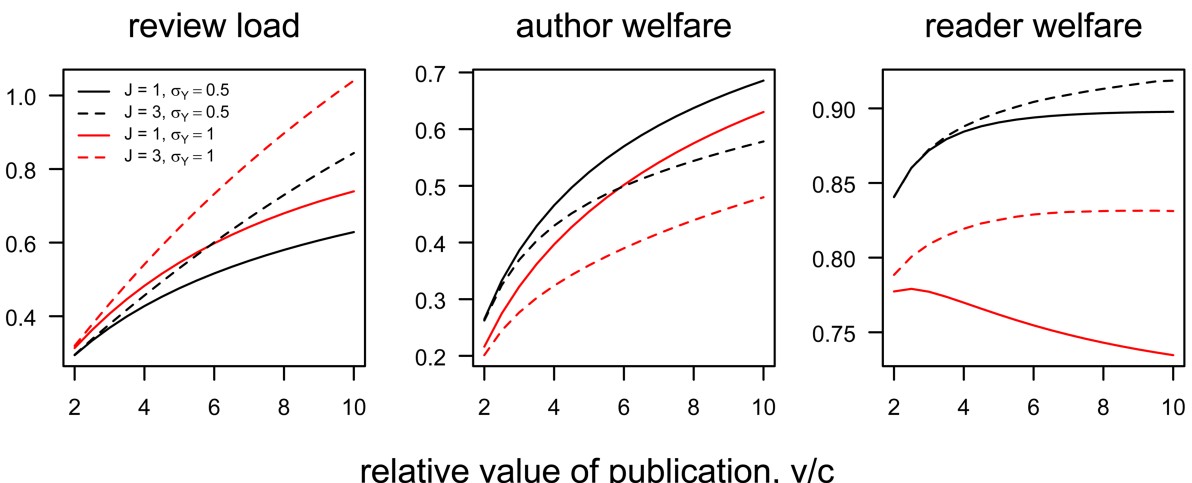

**Fig 5. Welfare consequences of journal proliferation.** Increasing the number of elite journals helps readers but hurts authors and reviewers, and these effects are amplified when peer reviewing is less accurate. In this figure, the review noise, $\sigma_Y$, is independent of the review load, $L$, to highlight how the number of journals, $J$, interacts with review noise. Left: Review load per period vs. $v/c$ for $J = 1$ (solid) or $J = 3$ (dashed) journals, with review noise $\sigma_Y = 0.5$ (black) or $\sigma_Y = 1$ (red). Center: Author welfare. Right: Reader welfare, equal to the (standardized) average value of a published manuscript. Throughout, $k = 0.2$ and $\sigma_X = 1$. Code to generate this figure can be found in https://zenodo.org/records/15866736.

$(\lim_{J\to\infty} m_J(q; y) = 1)$. Writing the limiting marginal author as $\hat{q}_\infty$, the capacity-filling condition becomes

$$k = \int_{\hat{q}_\infty}^1 dq = 1 - \hat{q}_\infty.$$

Thus $\hat{q}_\infty = 1 - k$.

In other words, when perfectly informed authors have an unlimited number of attempts to submit their work, screening is perfect: only the top $k$ authors ever submit their manuscript, and they keep submitting until their manuscript is published (Fig 2F). The *entire* function of peer review in this case is to drive the marginal author's probability of acceptance down to $c/v$ so that no other authors find it worthwhile to submit. The sorting function of peer review is just necessary waste, because every submitted manuscript is published eventually. Further, the equilibrium rejection rate increases with both $v/c$ and $\sigma_Y$. If $v/c$ rises, the acceptance threshold $\hat{y}_\infty$ must be pushed even higher to discourage non-submitting authors, leading to more superfluous rejection and resubmission of top papers. If $\sigma_Y$ increases, the acceptance probabilities of all authors above the marginal author decrease, forcing them to submit their paper more often before it is finally accepted.

### Desk rejection

As screening and sorting weaken, eventually a journal is bound to receive more manuscripts than it can review. Of course, journals are not obligated to send every manuscript out for review. Instead, journals can and do counter a surge in submissions by desk-rejecting some manuscripts, thus preserving their available review labor for the most promising submissions [49,50]. Yet anticipating the net effect of desk-rejection is complicated, because, as we have observed, authors choose whether or not to submit their manuscript in anticipation of the sorting process to follow. Presumably, even an informed journal editor will make less accurate decisions without peer reviews than with them, and so it is unclear on its face how desk rejection will affect screening, and how this will in turn affect the quality of a journal's published articles. Here, we add desk-rejection to the one-journal model to formalize this idea and briefly explore its potency. A formal model of desk-rejection with competing peer journals is beyond the scope of the present article, but we informally discuss how competition impinges on desk rejection later.

**A single journal.** To add desk-review to the model with a single elite journal, suppose that the journal editor observes a noisy signal $D$ of the manuscript's quality, with $D \sim N(\theta, \sigma_D^2)$. Journals rationally use a threshold rule to decide which papers to reject immediately and which to send out for peer review, with papers sent out for peer-review if $D \geq d$ for some threshold $d$. To keep matters simple, we assume that manuscripts sent out for review are accepted or rejected based on their review score $Y$ alone. Write author $q$'s acceptance probability when facing desk-rejection threshold $d$ and review threshold $y$ as $a(q; d, y) = \Pr\{D \geq d, Y \geq y | F_X(X) = q\}$.

For any possible desk-rejection threshold $d$, there is a corresponding marginal author $\hat{q}(d)$ and review score threshold $\hat{y}(d)$ that satisfy the author-rationality and capacity-filling conditions. The journal's utility $u(d)$ is the average quality of accepted manuscripts, which is given by the expression

$$u(d) = \frac{1}{k} \int_{\hat{q}(d)}^1 E\left[\theta | F_X(X) = q, D \geq d, Y \geq \hat{y}(d)\right] a(q; d, \hat{y}(d))\, dq.$$

At equilibrium the journal sets the desk-rejection cut-off $\hat{d}$ to maximize its utility:

$$\hat{d} \in \underset{d}{\operatorname{argmax}}\, u(d). \tag{JR}$$

We call this the *journal-rationality* condition. An equilibrium is given by a triple $(\hat{d}, \hat{q}, \hat{y})$ that solves the AR, CF, and JR conditions.

With desk rejection in play, use $S = 1 - \hat{q}$ to denote the volume of submitted manuscripts, and continue to use $L$ to denote the volume of manuscripts sent out for review. Author welfare is now given by $1 - c\,S/v\,k$, while the burden on reviewers continues to scale with $L$.

Fig 6 shows how desk-rejection affects the welfare of authors, readers, and reviewers of a single elite journal. These results show that, at least under the settings explored here, judicious desk-rejection benefits all parties. Readers (and the journal) benefit because the journal ultimately publishes better papers. Authors have fewer manuscript rejected and thus recoup more of the available surplus. Reviewers are faced with fewer manuscripts to review. Of course, the extent of these benefits depends on the accuracy of the editors' decisions, as captured by $\sigma_D$. As the accuracy of desk-review declines, its usefulness diminishes accordingly.

**Several competing journals.** A formal model that includes desk rejection with several competing journals lies outside the present scope, since it would require modeling how authors behave when they face journals with different desk-rejection cutoffs. However, intuition suggests that when competing journals rely on a common pool of reviewers, a "tragedy of the reviewer commons" [51] makes desk rejection less useful for managing the review load. The logic goes as follows. When a journal has its own private community of reviewers, it fully internalizes the cost of sending a manuscript our for review. At equilibrium, such a journal sets its desk-rejection policy to optimally trade off the cost of depleting its available review labor against the benefit of using peer review to learn more about a manuscript's quality. But when several competing journals rely on the same common pool of reviewers, then a journal does not fully internalize the cost of requesting a review, because the cost of depleting the available review labor is borne by all journals. Yet the requesting journal still captures the full benefit of obtaining a review and learning more about the manuscript under consideration. Thus, because journals that rely on a shared reviewer pool do not fully internalize the cost of requesting reviews, they will set a desk-rejection policy that desk-rejects manuscripts less often than would be socially optimal.

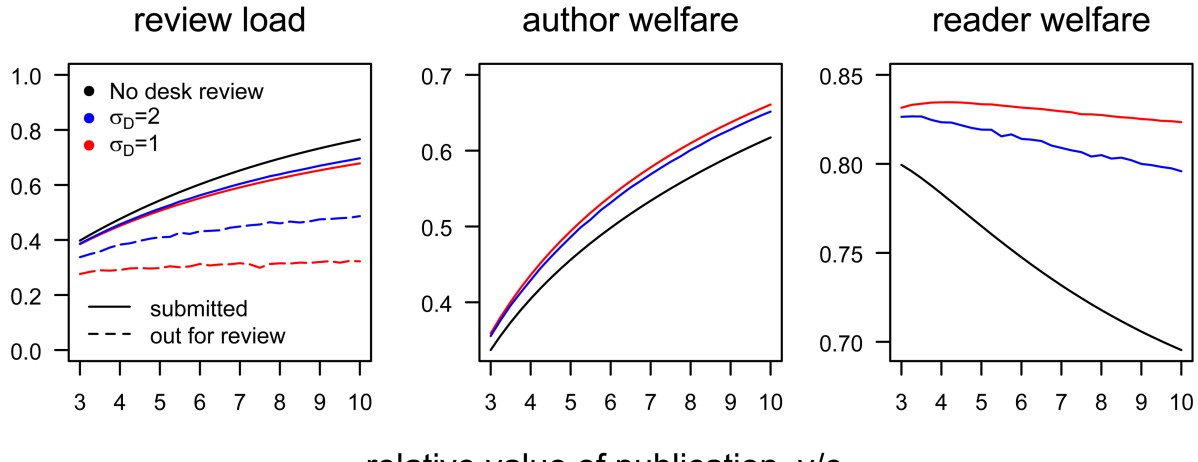

**Fig 6**. **Welfare consequences of desk rejection.** For a single journal operating in isolation, desk rejection makes all parties—reviewers, authors, and readers—better off. Welfare measures are shown when desk rejection is absent (black lines), noisy (blue, $\sigma_D = 2$), or more precise (red, $\sigma_D = 1$). Left: volume of manuscripts either submitted (solid lines) or sent our for review (dashed lines; submission volume and review load are the same without desk review). Center: Author welfare. Right: The readers' welfare, given by the (normalized) average quality of published manuscripts. For these results, $k = 0.2$, $\sigma_x = 1$, and $\sigma_Y = 0.25 + \sqrt{L}$. Code to generate this figure can be found in https://zenodo.org/records/15866736.

In the S1 Appendix, we formalize this idea with a simple mathematical model whose structure and logic parallel the classic Cournot oligopoly model in economics [52].

## Discussion

Some of the drivers of the peer-review crisis are straightforward and need little elaboration. The number of papers published has been growing about 5% annually since 1952 [53,54], a rate that exceeds the concomitant growth in the number of university faculty [55]. Of those faculty, a growing proportion are employed in short-term contingent positions and may be less able to devote time to unpaid and unrecognized review service. Rapid growth of scientific productivity in non-Western countries has outpaced the fraction of review invitations going to authors located in these regions [7], presumably because of mismatches in the composition of editorial boards. These trends have caused peer-review effort to become imbalanced, with a comparably small fraction of researchers providing most of the review labor [56,57].

Meanwhile, both the intrinsic rewards of reviewing and the cost of declining review invitations have shifted. Pre-print servers and their ilk have eliminated what used to be one of reviewing's major perks, namely the opportunity to obtain an early look at important unpublished work. Scientific communities are larger and looser knit, making the professional networks that bind editors and reviewers more diffuse. The universal reliance on e-mail for communication between journals and reviewers has made editors' invitations weaker signals of genuine interest in a reviewer's opinion compared to the manila envelopes and hand-written notes of yesteryear, while evolving social norms reduce the psychic cost to reviewers of ignoring e-mailed invitations altogether.

Other forces driving the peer-review crisis are subtle and more complex. We have argued here that the peer-review crisis is partially driven by a pernicious feedback loop in which a growing burden on the reviewer community leads to less accurate reviewing, which in turn compels authors to take more chances with regard to where they submit their work, which exacerbates the burden on the reviewer community further, and so on. Moreover, the decentralized nature of scientific publishing and the need for journals to compete for authors, readers, and reviewers short-circuits many of the most obvious paths towards halting or reversing this cycle. While we have presented our model in the context of elite journals, that focus is merely a simplification to streamline the exposition. Similar forces buffet journals across the scientific spectrum.

Like all models, our model makes simplifications that provide scope for future work. Perhaps the most substantively, we treat manuscript quality as exogenous. Of course, authors are not endowed with fully formed manuscripts, but instead they decide how much effort to invest to developing manuscripts based on their anticipation of the journal's scrutiny. For example, the rise of revise-and-resubmit as an author's likely best outcome may perversely encourage authors to submit manuscripts that are less than polished, both because revision seems inevitable and because journals can no longer expect polished submissions if unpolished submissions become the norm [58]. A model of journal and author behavior that endogenizes manuscript quality would provide more satisfying insight on this count.

If contemporary science has reached a point in which the demand for review labor outstrips the available supply, what (besides increasing desk rejection) might journals do to reconcile the two? Journals might consider tightening screening by increasing the cost of preparing a manuscript, $c$, thus discouraging more authors from submitting their manuscripts in the first place [2]. But monetary fees—whether charged upon submission or acceptance—are more likely to screen for authors with the institutional resources to pay [44], and costs generated by onerous submission requirements postpone the dissemination of results, thus making journals less marketable.

Moreover, competition for authors forces journals to keep their costs to authors low. Competing journals must simultaneously screen authors and compete for them, creating a complex set of incentives. While a proper treatment of the interaction between competition and screening behavior lies outside the present scope (but see ref. [44]), intuition suggests that competition for authors interferes with a journal's ability to screen. For example, if a journal increases its costs $c$, authors will face a trade off between the additional cost of submitting to the now costlier journal vs. the better odds of

acceptance at that journal resulting from thinner competition for publication slots. We conjecture that this trade-off will drive top authors who are insulated from the cost of thicker competition away from the costlier journal while attracting bottom authors to it. This adverse selection impels journals to keep their cost of submission low, as we observe throughout the current publishing environment.

If raising submission costs is off the table, journals might instead try to increase the supply of high-quality review labor. But journals that seek to recruit more high-quality reviewing are handcuffed by the tradition of peer review as a volunteer activity. In a typical labor market, employers can attract more labor by increasing wages. Journals operate without this basic market mechanism when reviewers are unpaid. If publishers were able to recruit more high-quality review labor by paying appropriate wages, they could maintain constant review quality in the face of an increase in submissions, breaking the feedback loop between screening and sorting.

To be sure, paying reviewers brings risks. Non-profit journals would need to pass the cost of reviewer pay on to authors or subscribers, although perhaps society journals could maintain a base of volunteer reviewers by cultivating a sense of shared responsibility for the journal. Paying reviewers also commodifies review labor, likely irreversibly [59]. This commodification could perversely decrease the availability of high-quality review labor if reviewers operating under the honor system perceive their effort to be worth more than the wages that the journal offers [26,59,60]. Nevertheless, recent experiments with modest payments to reviewers have succeeded in accelerating the pace of peer review, at least in the short run [23,25,26]. Wages needn't be simple direct payments, either. An intriguing alternative idea is to offer monetary prizes for top reviews, which could sweeten the pot for reviewers while motivating reviewer effort [61]. While substantial trial and error should be expected in establishing a plan that adequately compensates reviewers, once such a plan is established, it could give publishers a lever to recruit reviewers that they currently lack.

Are there other options that might salvage the institution of (voluntary) peer review? We highlight a few options here, all of which echo the suggestions of others. First, journals can reduce re-review of revised manuscripts [50,62]. Re-review is currently commonplace: accepted papers frequently go through at least two [63] or more [50] rounds of review prior to acceptance. But routinely sending revised manuscripts back out for review only burdens the reviewer community further while encouraging both authors and reviewers to modulate their effort in anticipation of the revision and re-review process to follow [58]. Journals can reduce re-review by empowering, or even encouraging, academic editors to be more proactive in their decision-making, especially when evaluating revised manuscripts. A more radical step to reducing re-review is to take revise-and-resubmit decisions off the table altogether [58]. Second, the scientific community can continue to explore ways of sharing reviews of rejected manuscripts, so that the associated review effort is not wasted. As our model with several journals suggests, much of the burden on the peer-review community flows from serial re-review of already rejected manuscripts when those manuscripts are sent to new journals. Cascading review and open-review platforms represent promising moves in this direction.

Finally, one last lever that our model suggests for salvaging peer review is to decrease $v$, the reward that authors reap for publishing in selective journals. In this model, we have taken the view that $v$ is set exogeneously by the scientific community and, unlike the other solutions we have considered, cannot be changed unilaterally by any single actor. Further, the forces that determine $v$ are varied and complex because the rewards attached to publication serve a variety of economic and sociological functions in science. For example, we [64] and others [65] have argued that rewarding publication constitutes a rational if imperfect scheme for motivating researchers to work hard and to take scientific risks while also protecting researchers' livelihoods from the vicissitudes of scientific chance. But this work does not consider the pressure that publication rewards place on peer review. Perhaps a more holistic view would suggest that the scientific endeavor would be better off if fewer rewards flowed to authors who publish in top outlets. Or perhaps it wouldn't; the answer remains far from clear. Regardless, whether science would be better off if the rewards to publication were reduced, and if so, how the scientific community might coordinate to change the prestige associated with publication seem to be rich and increasingly urgent topics to contemplate.

## Supporting information

**S1 Appendix. Mathematical proofs and additional results**
(PDF)

**S1 Fig. Optimal screening for maximizing reader welfare.**
(PDF)

## Acknowledgments

Having written a paper about the demise of peer review, we were heartened to receive three thoughtful, detailed, and on-point reviews of this manuscript. We thank the reviewers for their careful attention and constructive feedback, and for demonstrating that all is not yet lost. We also thank Kyle Myers and Nihar Shah for bringing AO's and Zhang et al.'s papers to our attention. KG thanks both the Department of Biology at the University of Washington and the Institute for Advanced Study in Toulouse for visitor support.

## LLM use statement

We used ChatGPT for coding assistance, spot-checking of numerical results, and to find some of the references herein. All of the text and mathematical results are our own human outputs.

## Author contributions

**Conceptualization:** Carl T. Bergstrom, Kevin Gross.

**Formal analysis:** Carl T. Bergstrom, Kevin Gross.

**Funding acquisition:** Carl T. Bergstrom, Kevin Gross.

**Writing – original draft:** Carl T. Bergstrom, Kevin Gross.

**Writing – review & editing:** Carl T. Bergstrom, Kevin Gross.

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
