## [Editor Report · Decision Letter 0]

15 Sep 2025

Dear Dr Gross,

Thank you for submitting your manuscript entitled "Will anyone review this paper? Screening, sorting, and the feedback cycles that imperil peer review" for consideration as a Meta-Research Article by PLOS Biology. Please accept my apologies for the very slow response.

Your manuscript has now been evaluated by the PLOS Biology editorial staff, and I'm writing to let you know that we would like to send your submission out for external peer review. I should tell you that we were unable to obtain advice from an Academic Editor in time, so we will be looking for some enthusiasm from the reviewers.

Once your full submission is complete, your paper will undergo a series of checks in preparation for peer review. After your manuscript has passed the checks it will be sent out for review. To provide the metadata for your submission, please Login to Editorial Manager (https://www.editorialmanager.com/pbiology) within two working days, i.e. by Sep 17 2025 11:59PM.

Kind regards,

Roli Roberts

Roland Roberts, PhD

Senior Editor

PLOS Biology

rroberts@plos.org

---

## [Decision Letter · Decision Letter 1]

3 Nov 2025

Dear Kevin,

Thank you for your patience while your manuscript "Will anyone review this paper? Screening, sorting, and the feedback cycles that imperil peer review" went through peer-review at PLOS Biology. Your manuscript has now been evaluated by the PLOS Biology editors, an Academic Editor with relevant expertise, and by three independent reviewers.

You'll see that reviewer #1 is broadly very positive about the study, but ends up wondering whether, for publication in PLOS Biology, you should do some further exploration around desk rejection behaviour. Reviewer #2 is also very positive, but asks about sensitivity to non-Gaussian distributions and the possibility of examining non-equilibrium scenarios; he also has a discussion point. Reviewer #3 is also enthusiastic, but thinks that you need to improve the accessibility of the paper for the typical PLOS Biology reader, making a number of helpful suggestions as to how this might be achieved.

IMPORTANT: I discussed the reviews and cross-comments between the reviewers with the Academic Editor. As a result, while the additional analyses requested by reviewers #1 and #2 would undoubtedly strengthen the study, we won't insist on them. I should also say that before review the Academic Editor had expressed some concerns somewhat similar to reviewer #3's final point: "Among possible remedies, they fail to mention the most obvious one: reducing the weight given to the number of publications and their JIF in assessing researchers. Nor do they discuss distributed review. The paper is elegantly written, and I agree the topic is of great interest to both researchers and publishers. But it offers neither group a practical way forward—especially if the arguments for particular remedies rest on equations that few people understand and even fewer could test." We think further consideration should be given to this.

In light of the reviews, which you will find at the end of this email, we are pleased to offer you the opportunity to address the comments from the reviewers in a revision that we anticipate should not take you very long. We will then assess your revised manuscript and your response to the reviewers' comments with our Academic Editor aiming to avoid further rounds of peer-review, although we might need to consult with the reviewers, depending on the nature of the revisions.

**IMPORTANT - SUBMITTING YOUR REVISION**

*Resubmission Checklist*

*Published Peer Review*

*PLOS Data Policy*

*Blot and Gel Data Policy*

Sincerely,

Roli

Roland Roberts, PhD

Senior Editor

PLOS Biology

rroberts@plos.org

REVIEWERS' COMMENTS:

Reviewer #1:

[identifies himself as Luis A. Nunes Amaral]

Bergstrom and Gross study the vicious feedback cycles of increased submission rates and less qualified peer review of manuscript. Their work follows along the lines of work by Adda and Ottaviani (2023) on grant awarding and Zhang et al (2023) on CS conference peer review but provides a clear improvement on those works in terms of both readability and parsimony.

The basic idea of the study, as I see it, is that the existence of highly-selective journals with a disproportionally high reward $v$ compared to publication in other venues (in the study modeled as a mega journal) leads to a peer-review pressure on the selective journal that results in reviewer scarcity, lower quality reviews, and incentives for submission of less worthy manuscripts to the selective journal in a vicious cycle of overburdened reviewers and editors.

Interestingly, the study also shows that the feedback loop between review load (L) and review quality ($\sigma_Y$) leads to prolonged transient perturbation away from equilibrium due to external shocks.

Importantly, the study also shows that the existence of multiple selective journals aggravates this issue by contributing to the lowering of the quality of peer review (measured by increased $\sigma_Y$).

Finally, the authors explore the impact of desk rejection on reviewer load and submission rate. The focus on a single selective journal and assume, reasonably, that editor review quality is likely lower than author's or peer reviewers'. Despite this, they show that desk rejection increases everyone's welfare: reviewers, authors and readers. This has to be somewhat reassuring for all of us involved in the business of science.

Nonetheless, Bergstrom and Gross note that when there are multiple selective journals, the editors will tend to set desk-rejection rates that are lower than what would be socially optimal.

This is clearly a manuscript on a very important matters. It is also a very readable manuscript that will almost certainly be more impactful that the 2023 studies mentioned earlier. The question, however, is whether it should be published in PLOS Biology, itself an elite journal?

This is a difficult question for me to answer. My inclination is to suggest that the manuscript needs more analyses is order to justify publication in PLOS Biology.

The authors explicitly state that: "Vertical" proliferation, in which journals more densely occupy niches along a prestige spectrum, is outside the present scope (but see ref. [38]).

I accept this choice even though my lab's research demonstrated the existence of such vertical integration (Stringer et al PLOS One and JASIST).

Nonetheless there is an interesting mechanism that is not explored but is definitely important. Again, this is inspired by my lab's research (Moreira et al PLOS One). It seems plausible that authors have a specific $\bar{q}$. That is, a manuscript $i$ produced by author j does not have $q_j^i$ drawn from $N(0, \sigma_x)$ but instead from $N(\bar{q_j}, f*\sigma_x$ where $f < 1$. This fact may not appear to be important for the default case but I think it is important for the desk review case. While the editor's $\sigma_D$ may be large, one could imagine that over time an editor will learn to recognize high $\bar{q_j}$ authors and be guided by it for deciding on desk rejection.

I believe that an exploration of the impact of such a mechanism on desk rejections in terms of societal welfare would be important, especially in terms of exclusion of high $\bar{q_j}$ authors early in their career and slacking of high $\bar{q_j}$ authors late in their career.

Reviewer #2:

[identifies himself as Stefano Allesina]

First of all, I would like to respond to the manuscript title in the affirmative. This is a very interesting and thought-provoking manuscript: the Authors adapt and extend a previously-published model for peer-review and use it to illustrate some of the problems that plague the system. The manuscript is well-written (I found only one typo, l486) and easy to follow. I have to offer only two somewhat technical comments, and some reflections.

The model is deeply rooted in normal distributions: the manuscript quality, the errors in evaluating the quality by both authors and reviewers are all Gaussian. While this makes a lot of sense when modeling errors, other quantities (esp. manuscript quality) could have very skewed distributions. For example, the number of citations, which can be considered a first-order estimate of readership, tends to be lognormally distributed. If I am not mistaken, changing the distribution of manuscript quality would have absolutely no effect on the conclusions (because only the percentile of quality matters). If this is the case, and whenever this is the case, the Authors could add a line saying that these assumptions do not matter (at least qualitatively). In a similar vein, one drawback of Gaussians is that the error functions cannot be expressed algebraically, and as such for example the existence of an equilibrium can be proven, but cannot be written explicitly in closed-form using the model parameters. Would the choice of different distributions alleviate this issue? If one had an explicit formula for \hat{q} and \hat{y} in terms of v/c and the various standard deviation, it would be very easy to show the effect on welfare etc by taking derivatives.

The analysis is very much based on equilibria: the existence of critical values \hat{q} and \hat{y} is proven, and then these values determine the welfare for authors, readers and reviewers. While reading the manuscript I kept thinking that a) authors have but a vague idea of \hat{y}, and b) journals also have only a rough estimate of \hat{q}. In reality, both authors and journals constantly adjust their submission/acceptance criteria based on a rapidly-shifting landscape, the birth/death of journals, etc. The analysis, as it stands, reminds me of classic game theory: the authors try to maximize their payoff, and journals to fill their pages with the best manuscripts, all seen as a "static game". I imagine that the same could be turned into an "evolutionary game", in which journals and authors correct their course in time, as a result of previous outcomes. A dynamic version of the model would also allow to study the relaxation to equilibrium following a perturbation (e.g., the cited increase in submissions following the pandemic), as well as the study of "interventions" on the parameters of the game. Right now, a dynamic version is mentioned in passing (l236), but in my opinion it would be an exciting avenue for the future.

In terms of reflections, I appreciated the thoughtful discussion, which is very much centered on the cost-side (c) and on increasing the supply of reviewers (for example by paying them). One aspect that I did not find discussed, but that greatly impressed me over the past 20 years, is the rise of bean-counting institutions (funding agencies, tenure committees etc) that want to measure the value (v) discussed in the manuscript directly (as impact factor, classification of journals into tiers, etc), and use it for real-world decisions. Even though we know that all Deans can count, but few can read, scientists have a role to play in reducing the differential value of publication venues for careers, and increase the weight attributed to the quality of the work (as opposed the impact factor of the journal in which it was published). For example, my institution limits the number of publications that can be submitted in a tenure/promotion package to five, thereby incentivizing researchers to concentrate on the quality of their work, rather than the number of papers published.

In summary, I very much appreciated the focus on a bare-bone, tractable model of peer-review that, by removing some of the complications and focusing on the main processes, can recapitulate the sorry state of the peer-review system and shed light on the major levers one can act upon to improve it.

Reviewer #3:

[identifies himself as Stephen Curry]

This is a very interesting paper which offers a mathematical model of the widely appreciated problem of the increasing burden of peer review. The primary model explored considers a world in which there is just one elite journal and one 'mega-journal' and is, by the authors' own admission, relatively simplistic. This model is later elaborated to consider publishing landscape in which there are multiple 'elite' journals (of equal standing), and then also considers the impact of desk rejection. Although I would judge myself reasonably well versed in mathematics, I cannot claim sufficient understanding to pronounce confidently on the rigour of the mathematical models developed here. I made a decent fist of following the derivation of the single elite journal model, but I struggled with the more elaborate attempts to incorporate multiple elite journals and desk rejection. I trust that other reviewers will bring greater expertise to this task.

That said, I guess I might reasonably be considered a not untypical reader of PLoS Biology. It seemed to me that the single elite journal model, which forms the bulk of the manuscript, was reasonable despite its simplicity and provided insights on the feedback effects between author screening and journal sorting that comprise a rational explanation of the growing burden of journal peer review. I am left wondering, nevertheless, if my positive impression of this part of the paper is an example of confirmation bias. This feeling is more acute for the more elaborate models that I understand less well mathematically.

Nevertheless, in my view the paper is an interesting contribution to an important discussion. I would like to see it published so that readers can judge for themselves the merit and power of the approach taken by the authors. In its present state, I think there is a danger that many readers seeing the simplifying assumptions of the models might too quickly disregard the insights they have to offer, so some additional effort should be made in the presentation of the work, both to explain the workings of the model to those less conversant with the mathematical techniques used and to emphasise how, despite the simplifying assumptions, the model relates to the real world of modern day scholarly publishing. I have some specific comments along these lines that, if addressed, may give the paper greater impact with its intended audience. They are a mixture of major and minor comments. I will lay them out in the order that they occur in the manuscript.

Line 2 - a minor point but the claim that publishers rely on peer review to identify 'the best science' is over-simplistic and starts the paper off, if not on the wrong foot, then with a stumble. There are countless mid and lower tier journals where manuscripts reporting perfectly decent science are submitted but which neither the authors not the reviewers would consider 'the best science'. The best submissions for a given journal might be a better description; or perhaps it could be claimed that peer review strives to provide a valuable quality check on submitted manuscripts (its well-known weaknesses notwithstanding).

Line 27 - another minor point but I doubt that many authors have 'full awareness' of the evaluation to follow. I would qualify slightly - probably they have a reasonable idea of the standard expected of their journal of choice.

Line 62 - this reader (as I suspect many other readers of PLoS Biology) is not familiar with the literature on agent-based models. It would be helpful to have a brief explanation of what these are and how they differ from the approach taken by the authors. I was left wondering why agent-based models were unable to unearth the feedback mechanisms that are a key - novel? - feature of Bergstrom & Gross's model.

Page 4 - Fig 1 - The legend refers to screening but not sorting. While these terms are defined implicitly in the body text, it would be useful to give an explicit definition and use the terms consistently throughout.

Pages 4-6 - the description of the mathematical model is reasonably clear but would benefit from some additional explanation for a wider readership. Line 88 - 'journal capacity' is pretty self-evident but the reader will have to work less hard if it is also stated that this means the total number of articles published (per issue or per year). Line 93 - explain why it helps to suppose that there is a 'unit mass of authors'. Line 94 - why 'is endowed with'; why not the simpler 'has'? Lines 96 and 98 mention the authors "own sense of whether their work is any good" and "their private signal X" of their manuscript's quality; wouldn't it be better not to have two different descriptions of the same thing? Line 100 - explain what is meant by 'quantile'.

Line 120 - I did not understand the sentence in parenthesis about indifferent authors.

Page 7 - Fig. 2 - I think these graphs would be clearer if they had more annotation (to reduce the need to consult the legend); e.g. add the values of sigma or v/c to the plot ((e.g. in panel A, add v/c = 5 etc). If panels B & C could be side by side, that would facilitate the comparison discussed in the text.

Line 159 - In addition to presenting the expression for authors' welfare, I would suggest saying in words that the welfare diminishes with c and L, and increases with c and k - just for maximum explicitness.

Line 165 - the statement "The readers' welfare measures how reliably publication indicates scientific quality" is indicative of the simplicity of the model since, arguably, the quality of the manuscripts in any given journal takes a range of values. Perhaps some qualification of the statement could be offered or a less definitive verb than "measures" could be used? "Estimates"?

Line 172 - typo? "costs of reviewing" rather than "costs to"?

Line 196 - typo? "analyses" plural?

Page 10 - Fig.3 - It is difficult to discern the red-blue gradations of the points in the plot. Might a graded, multi-colour scheme be easier to discern?

Line 214 - replace (or supplement) "reviewing" with "sorting" for greater consistency?

Line 224 - "Figure 4A, B illustrates" is not a sentence since there is no object.

Lines 229-234 - The explanation given is not very clear. Some tweaks are needed; e.g. "when reviewer accuracy declines as L increases" is better than "when reviewer accuracy declines with L" in this reader's opinion.

Lines 254-256 - why is vertical proliferation outside the scope of the paper? It strikes me as a more interesting question than horizontal proliferation? Can the authors explain?

Line 298 - I struggled to understand mathematically what "proportional to (the opposite of)" means. Presumably the relationship is not an inverse proportionality, but can the authors explain? Might it simpler and clearer to reframe things in terms of reviewer burden rather than reviewer welfare?

Line 334 - "writes neatly as"? Would "can be written neatly as" be more grammatically correct?

Line 435 - the authors may well be correct, but the use of the word "psychic" conjured for me a group gathered around a table in a darkened room for a séance! Might "psychological" be a good substitute?

Line 450 - revise-and-resubmit options might encourage authors to submit less polished manuscript but in my experience the rise of preprints opposes this tendency since any manuscript defects are on public display from the get-go.

Line 477 - "Top authors benefit from accurate reviewing and thus prefer to submit their work to journals that offer it." strikes me as a not unreasonable statement given the simplifying assumptions of the model, but what do authors know about the quality of review at different journals in reality (beyond, obviously, the extremes). Attempts to relate the model to reality would benefit from a more informed consideration of this point and citation of studies that have attempted to evaluate variation in the quality of peer review at different journals.

Discussion section - A major weakness of the discussion of the peer review problems that are the focus of this manuscript is the fact that the authors only consider what journals might do. This narrowness of perspective arises naturally from the construction of the model which centres on the triad of authors, reviewers and readers brought together by the journal. But to give the paper greater relevance, the discussion should be widened out. How might other stakeholders - e.g. universities, learned societies, funders- contribute to solving the problem? What about the capture of peer review by commercial publishers, who have sought to increase market share through journal proliferation? I appreciate that the authors will not want to provide a review of all the woes of academic publishing, but some discussion of how their model might help prompt action from stakeholders beyond journals or publishers, or might justify new or more radical approaches to scholarly communication, would be valuable.

---

## [Editor Report · Decision Letter 2]

22 Jan 2026

Dear Dr Gross,

Thank you for your patience while we considered your revised manuscript "Will anyone review this paper? Screening, sorting, and the feedback cycles that imperil peer review" for publication as a Meta-Research Article at PLOS Biology. This revised version of your manuscript has been evaluated by the PLOS Biology editors and the Academic Editor.

Based on our Academic Editor's assessment of your revision, we are likely to accept this manuscript for publication, provided you satisfactorily address the following data and other policy-related requests.

a) We need your Title to be more explicit and to lack punctuation. We suggest changing it to something like, "Identification of the threats to the sustainability of the peer-review system and possible mitigation strategies" (ideally we'd also want an active verb, but we can live without that...)

b) Many thanks for providing the underlying R code. Please cite the location of the data clearly in all relevant main and supplementary Figure legends (i.e. Figs 2-5, S1), e.g. “The code underlying this Figure can be found in https://zenodo.org/records/15866736”

We expect to receive your revised manuscript within two weeks.

*Published Peer Review History*

*Press*

Sincerely,

Roli Roberts

Roland Roberts, PhD

Senior Editor

rroberts@plos.org

PLOS Biology

---

## [Editor Report · Decision Letter 3]

28 Jan 2026

Dear Kevin,

Thank you for the submission of your revised Meta-Research Article "Screening, sorting, and the feedback cycles that imperil peer review" for publication in PLOS Biology. On behalf of my colleagues and the Academic Editor, Ulrich Dirnagl, I'm pleased to say that we can in principle accept your manuscript for publication, provided you address any remaining formatting and reporting issues. These will be detailed in an email you should receive within 2-3 business days from our colleagues in the journal operations team; no action is required from you until then. Please note that we will not be able to formally accept your manuscript and schedule it for publication until you have completed any requested changes.

Sincerely,

Roli

Senior Editor

PLOS Biology

rroberts@plos.org